:◉: PLOS | ONE

# Abundance of ethnically biased microsatellites in human gene regions

Nick Kinney[ID][1,2¤]*, Lin Kang[1,2], Laurel Eckstrand[3], Arichanah Pulenthiran[ID][1], Peter Samuel[ID][1], Ramu Anandakrishnan[ID][1], Robin T. Varghese[ID][1], P. Michalak[1,3,4], Harold R. Garner[1,2]

**1** Edward Via College of Osteopathic Medicine, Blacksburg, VA, United States of America, **2** Gibbs Cancer Center & Research Institute, Spartanburg, SC, United States of America, **3** Virginia-Maryland College of Veterinary Medicine, Blacksburg, VA, United States of America, **4** Institute of Evolution, University of Haifa, Haifa, Israel

¤ Current address: Primary Care Research Network and the Center for Bioinformatics and Genetics, Edward Via College of Osteopathic Medicine, Blacksburg, VA, United States of America
* nkinney@vcom.vt.edu

**Data Availability Statement:** All relevant data are within the manuscript and its Supporting Information files.

## Abstract

Microsatellites–a type of short tandem repeat (STR)–have been used for decades as putatively neutral markers to study the genetic structure of diverse human populations. However, recent studies have demonstrated that some microsatellites contribute to gene expression, *cis* heritability, and phenotype. As a corollary, some microsatellites may contribute to differential gene expression and RNA/protein structure stability in distinct human populations. To test this hypothesis, we investigate genotype frequencies, functional relevance, and adaptive potential of microsatellites in five super-populations (ethnicities) drawn from the 1000 Genomes Project. We discover 3,984 ethnically-biased microsatellite loci (EBML); for each EBML at least one ethnicity has genotype frequencies statistically different from the remaining four. South Asian, East Asian, European, and American EBML show significant overlap; on the contrary, the set of African EBML is mostly unique. We cross-reference the 3,984 EBML with 2,060 previously identified expression STRs (eSTRs); repeats known to affect gene expression (64 total) are over-represented. The most significant pathway enrichments are those associated with the matrisome: a broad collection of genes encoding the extracellular matrix and its associated proteins. At least 14 of the EBML have established links to human disease. Analysis of the 3,984 EBML with respect to known selective sweep regions in the genome shows that allelic variation in some of them is likely associated with adaptive evolution.

## Introduction

Approximately two thirds of the human genome consists of repetitive DNA [1]. These repeats vary in size, complexity, and abundance in the genome: microsatellites are perhaps the simplest. Each microsatellite consist of a short motif (1–6 base pairs) repeated in tandem to form an array [2]; over 600,000 unique microsatellites exist in the human genome [3, 4]. Despite the

**Funding:** This work was funded by a grant from the Bradley Engineering Foundation to the Edward Via College of Osteopathic Medicine and by a grant from Edward Via College of Osteopathic Medicine to NAK. HRG is the founder and co-owner of Orbit Genomics and received support in the form of salary. The funders had no role in study design, data collection and analysis, decision to publish, or preparation of the manuscript.

**Competing interests:** HRG is the founder and co-owner of Orbit Genomics which may be interested in licensing these findings. Orbit Genomics was not involved with any aspect or in funding of this research. This does not alter our adherence to PLOS ONE policies on sharing data and materials. There are no patents, products in development or marketed products associated with this research to declare.

simplicity of microsatellites, they have been leveraged in forensic and kinship analysis for decades. Essentially, they serve as genetic fingerprints; a consequence of their high mutation rate. In addition, microsatellites have a well-established role in diseases such as fragile X syndrome, spinocerebellar ataxias, myotonic dystrophy, Friedrich ataxia, and Huntington's disease [5, 6].

Recently, microsatellites have garnered interest for their role in complex diseases and subtler effects on gene expression [7–10]. Variations in the length of repeat arrays influence gene expression by inducing Z-DNA and H-DNA folding [10]; altering nucleosome positioning [10, 11]; and changing the spacing of DNA binding sites [2, 9, 12]. In fact, a recent genome wide survey of short tandem repeats (STR) identified 2,060 that affect nearby gene expression (eSTRs) and estimated that STRs contribute up to 15% of the *cis* heritability among all types of genetic variants [9]. Strong enrichments were found near transcription start sites and predicted enhancers [9]. Shortly before this work, another study concluded that microsatellites facilitated divergence of gene expression in humans and great apes [13]. Thus, microsatellites have the capacity to affect gene expression and may be leveraged by natural selection for efficient evolution. A review published on the heels of both studies suggested that microsatellites contribute to the missing heritability of polygenic disorders and called for a better understanding of the repeatome at large [8].

Despite our nascent understanding of microsatellite function, they are well-studied in diverse human populations. Landmark studies in the 1990's showed that microsatellites can be used to infer the demographic history of human populations [14–17]. In particular, a 1992 study of four racial/ethnic groups–African American, Mexican American, Asian, and white–concluded that microsatellites can be leveraged for individual identification [18]. This idea was implemented in a 1994 study that succeeded in clustering individuals according to their geographic origin [19]: microsatellite diversity was highest in Africans. Subsequent studies established that only a modest number of microsatellites are required to make reliable inferences and set precedents for a 2003 study of 377 microsatellites in 52 worldwide populations [20–22]. A more recent study used microsatellites to characterize genetic variation across 121 African populations, four African American populations, and 60 non-African populations [23]. Genetic diversity was shown to decline with distance from Africa; private alleles were more numerous in Africa than in other regions; and 14 ancestral population clusters were revealed [23].

Microsatellites are now routinely used to study the genetic structure of diverse human populations [24–28]. Informative panels range from ten to several thousand loci and often draw from the Marshfield screening sets [29]. One of the largest studies to date merged 8 datasets [30]: the aggregated data included 645 microsatellite loci with genotypes in 5,795 individuals from seven population groups. Africans, East Asians, Oceanians, and Native Americans formed distinct clusters; Europeans and South Asians formed part of a central heterogeneous cluster [30]. This overall pattern was reiterated in a smaller study of 46 ancestry informative markers [31]; principal component analysis (PCA) revealed distinct clusters for Africans and East Asians with overlapping central clusters corresponding to South Asians and Europeans, respectively [31]. Similar results have been obtained with samples drawn from the 1000 Genomes Project Phase 1 demonstrating the utility of microsatellite analysis from Illumina sequencing [32].

We use existing sequencing data from the 1000 Genomes Project Phase 3 to build on what is known about polymorphic germline microsatellites. First, we preform PCA of 316,147 microsatellites: a significant fraction of all microsatellites in the Human Genome. The pattern of variation is consistent with previous studies that focus exclusively on smaller sets of polymorphic microsatellites [30–32]. We use Fisher's exact test to identify 3,984 ethnically-biased

microsatellite loci (EBML); for each EBML at least one ethnicity has genotype frequencies statistically different from the remaining four. We find significant overlap between EBML and previously identified eSTRs: a key result of this study. EBML are enriched with core matrisome and matrisome associated genes. Further enrichments are found when we characterize EBML with respect to introns, exons, UTRs, and coding sequences (cds). EBML are over-represented in soft sweep regions of the human genome suggesting that microsatellites contribute to differential gene expression in distinct populations and have adaptive potential.

## Results

### Abundance of EBML in the human genome

Numerous studies have used microsatellites to reveal patterns of genetic variation within and between diverse human populations [23–28]; marker panels typically include ten to several thousand loci. We use next-generation sequencing (Illumina) to investigate 316,147 genome wide microsatellites in 2,529 samples belonging to five super-populations (ethnicities). PCA clusters reiterate the five ethnicities (**Fig 1A**) confirming the utility of microsatellite analysis

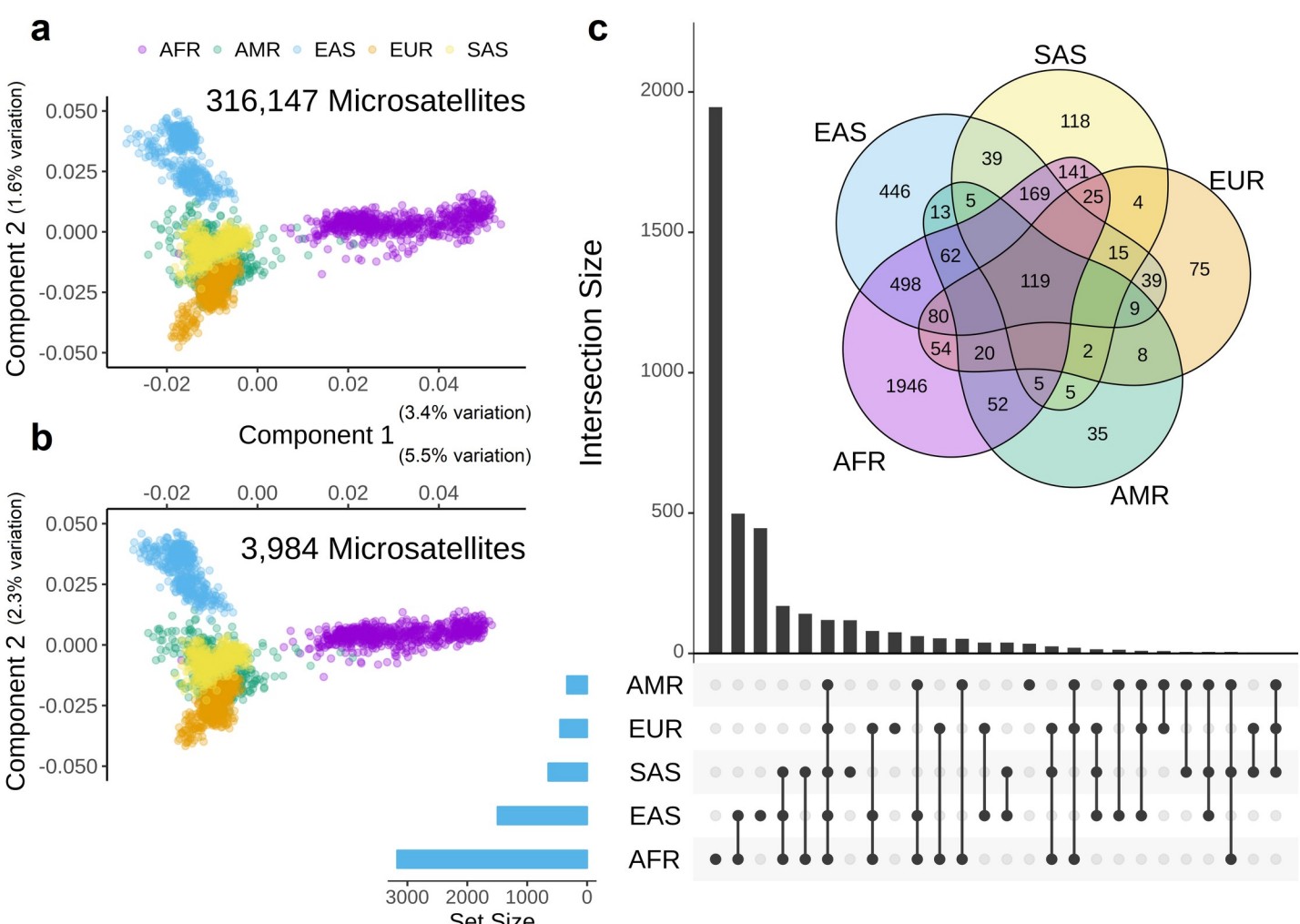

**Fig 1. Discovery and analysis of EBML. a,** Principal component analysis of 316,147 genome wide microsatellites reveals a distinct pattern of variation in 5 super-populations (ethnicities). **b,** Principal component analysis of 3,984 EBML reveals a pattern of variation similar to genome wide microsatellites. **c,** Number of EBML identified in five ethnicities: UpSet plot inset with 5-way Venn Diagram. Overlap indicates microsatellites specific to two or more ethnicities. Oval regions sum to 3,171 (AFR); 1,494 (EAS); 450 (EUR); 647 (SAS); and 335 (AMR). All regions sum to 3,984: the total number of EBML identified in this study.

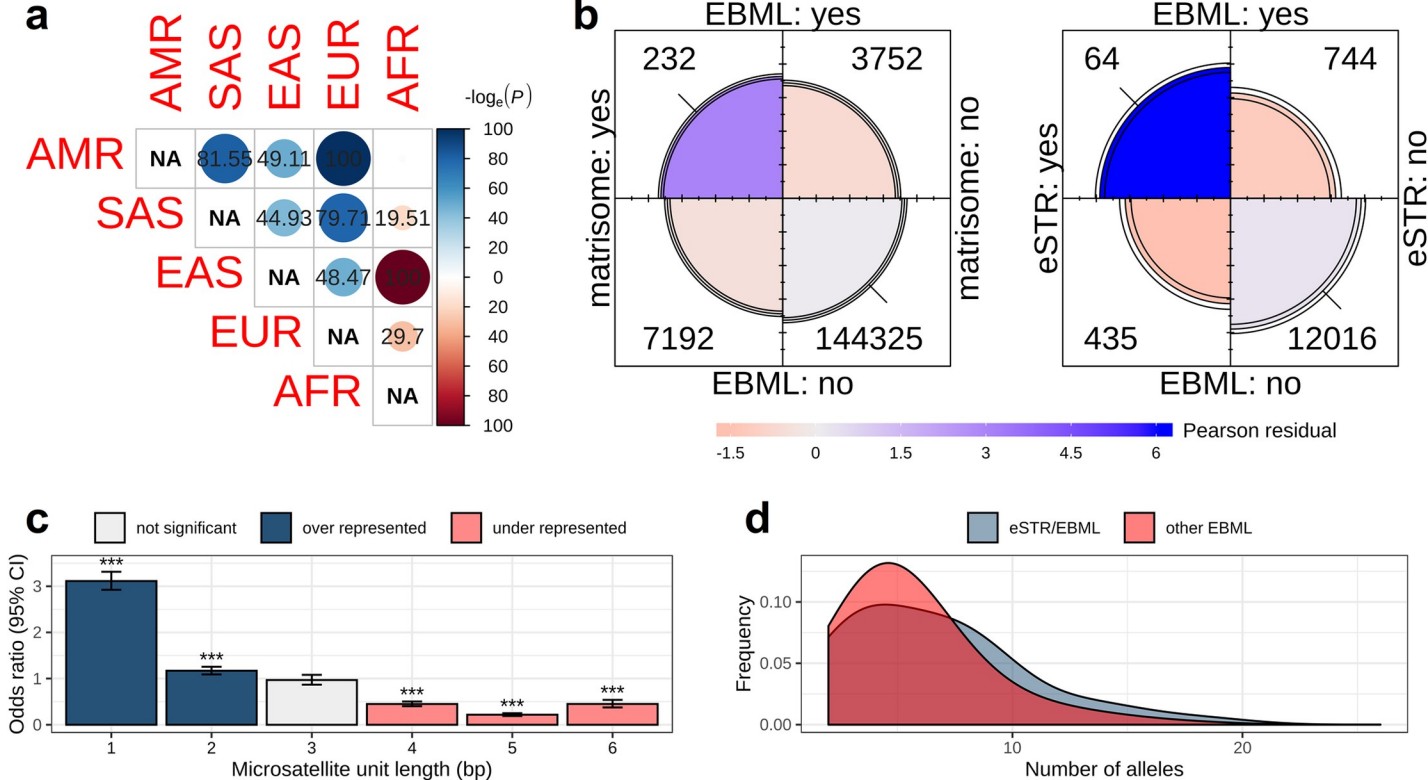

**Fig 2. Summary of significant enrichments in the set of 3,984 EBML. a,** EBML for each of the 10 super-population pairs were used to construct a 2x2 contingency table followed by $\chi^2$ test of independence: matrix entries shown as -log(p-value). Each matrix entry corresponds to a test of the null: that EBML are independent in the pair of super-populations. Over-representation shown in blue; under-representation shown in red. With the exception of AFR, microsatellites specific to two or more super-populations are over-represented. **b,** Fourfold plots of EBML reveal significant overlap with matrisome genes (left) and eSTRs (right). Area of each quarter circle is proportional to the cell frequency (following marginal standardization). Color of each quarter circle corresponds with its Pearson residual: blue indicates the cell entry exceeds the expected value; red indicates less than then expected value. Confidence rings for the odds ratio allow a visual test of the null (no association); here, 99% confidence rings do not overlap indicating the null is rejected. **c,** EBML are enriched with 1-mer and 2-mer repeats. Enrichments for each unit length were checked by constructing a 2x2 contingency table followed by $\chi^2$ test of independence. Bars show the odds ratio with 95% confidence interval: over-represented motifs shown in blue; under-represented motifs shown in red. Level of significance (p-value) is indicated symbolically: $p<0.05$ (*); $p<0.01$ (**); $p<0.001$ (***). **d,** The 64 EBML known to affect gene expression (eSTRs) have more alleles on average than the remaining 3,920 EBML; smoothed distribution of allele counts shown in blue and red, respectively. The difference is statistically significant (p = 0.01; two sided Kolmogorov-Smirnov test).

from Illumina sequencing [32]. The overall pattern of variation is consistent with previous studies that focus on smaller panels of known polymorphic microsatellites [30–32]: AFR, EAS, and EUR in three outside clusters with AMR and SAS in two overlapping central clusters (**Fig 1A**). We use Fisher's exact test (see methods) to identify 3,984 ethnically-biased microsatellite loci (EBML); for each EBML at least one ethnicity has genotype frequencies statistically different from the remaining four. With the exception of AFR, EBML show significant overlap in all ethnicity pairs (**Fig 2A**). On the contrary, microsatellites specific to AFR are under-represented in SAS, EAS, and EUR. PCA of the 3,984 EBML reveals a pattern of variation representative of all 316,147 genome wide microsatellites (**Fig 1B**). The first two principal components suggest that variation is greatest in AFR and EAS, respectively. Once again, this finding is consistent with previous studies [33]; and, is recapitulated in all subsequent analysis (see results below).

Identifying EBML is an easier task than explaining their origin. EBML could emerge due to high microsatellite mutability, genetic drift, isolation-by-distance, or natural selection. The action of natural selection–investigated in the next section–would suggest adaptive potential.

Regardless, the 3,984 EBML add to the known polymorphic microsatellites and could be useful in future studies of population structure; a complete summary is shown in (**Fig 1C**) (**S1–S6 Tables; S1 Code; S1 Dataset**).

## Correspondence between EBML, eSTRs, and selective sweeps suggest adaptive potential

A recent study used RNA sequencing of lymphoblastoid cell lines to investigate links between array length variations in 80,980 short tandem repeats (STRs) and expression of nearby genes [9]. The study identified 2,060 significant associations (among the 80,980) which established the importance of expression STRs (eSTRs). Cross-referencing the 80,980 STRs against our 316,147 microsatellites reveals 13,259 repeats in common. We constructed a 2x2 contingency table based on classifications (eSTR and/or EBML) of the 13,259 shared repeats (**Fig 2B; right panel**). Remarkably, 64 loci classify as an eSTR and EBML (S7 Table); the overlap is statistically significant (p = 1.53e-8; $\chi^2$ test). Interestingly, the overlapping set of 64 EBML/eSTRs average 6.63±3.86 alleles each (424 alleles total) while the remaining 3,920 EBML average 5.68±3.03 alleles each (22,278 alleles total); the difference is statistically significant (p = .011; Kolmogorov-Smirnov test) (**Fig 2D**). The set of 64 includes all five ethnicities: 53 AFR, 32 EAS, 17 SAS, 16 EUR, and 9 AMR. In addition, five are biased in every ethnicity and five are embedded in coding sequences. Genes harboring the former set of five include SNX2 (3'-UTR), CST3 (intron), C2orf50 (5'-UTR), ACP2 (5'-UTR), and VEGFB (intron); the five coding microsatellites are in NOP9, USP36, PTPN18, AK9, and SNAPC4.

The correspondence between EBML and eSTRs suggest some microsatellites contribute to differential gene expression; however, this does not necessarily imply they have adaptive potential. To infer adaptive potential we identify microsatellites in selective sweep regions of the human genome. Briefly, selective sweep occurs when strong positive selection–due to a novel allele–reduces nearby genetic variation; sweep regions have been established for six populations in the 1000 Genomes Project [34]. We find 434 (out of 3,984) EBML in soft sweep regions; tested microsatellites have 30,850 (out of 316,147). The difference is small but statistically significant (p = 0.018). The 434 EBML include 21 in the coding sequence of 18 genes (**S9 Table**); one is a previously identified eSTR (CDS of gene *USP36*).

Overall, these findings suggest a degree of mutual overlap between EBML, eSTRs, and selective sweeps in the human genome. The interpretation of this overlap is built up from the definition of each region. Briefly, each eSTR has the capacity to affect gene expression; and, for each EBML at least one ethnicity has a distribution of genotypes statistically different from the remaining four. Thus, any correspondence between the two implies differential gene expression in one or more human super-populations (ethnicities). These differences likely stem from a complex combination of high mutability, genetic drift, isolation-by-distance, and natural selection. Drift, mutability, and isolation are inevitable; but, the situation is less clear for natural selection. The correspondence between EBML and selective sweeps suggests they have adaptive potential and may be targeted by natural selection [11, 35–38]. On the other hand, it is possible that EBML are simply in linkage disequilibrium with targets of selection such as nearby SNPs. Overall it seems likely that some EBML–particularly the ones that overlap eSTRs–have bona fide adaptive potential.

## Introns are over-represented among EBML; coding regions are under-represented

We characterize the 3,984 EBML in terms of their overlap with 3,324 introns, 342 exons, 542 UTRs, and 147 coding sequences. Although introns are the majority among tested

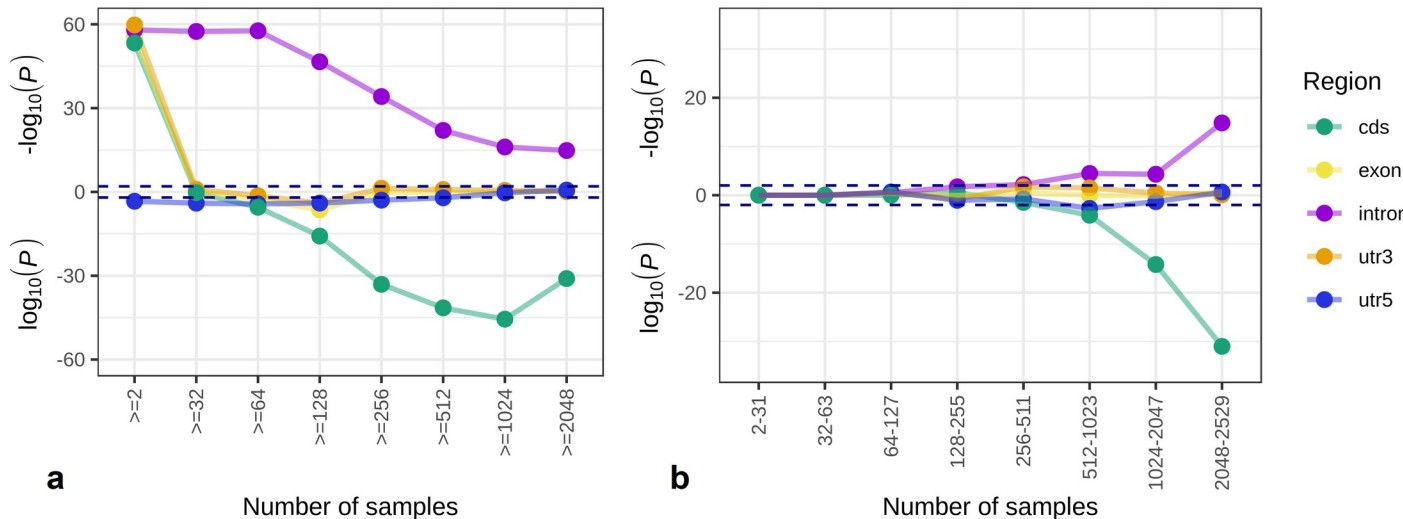

**Fig 3. Enrichment analysis of EBML with respect to gene regions: Introns, exons, coding sequences, and UTRs.** Analysis of microsatellites draws upon whole exome sequenced samples from the 1000 Genomes Project; consequently, enrichment with respect to gene regions could not be checked by directly comparing all 316,147 tested microsatellites to the 3,984 EBML. Instead, two series of enrichment tests are performed on subsets of the tested and EBML. **a,** In the first series, each iteration removes microsatellites based on the minimum number of available samples. **b,** In the second series, each iteration considers microsatellites within a range of available samples. Each point represents a $\chi^2$ test of the null: no association between EBML and the gene region. Over represented regions are plotted as -log(p) and appear above the x-axis; under represented regions are plotted as log(p) and appear below the x axis. Statistical significance (p = .05) is indicated with a dashed line. The null is rejected for intronic microsatellites (over-represented) and coding microsatellites (under-represented). The same conclusion is reached in both panels suggesting the results are robust to details of the analysis. Note: exonic microsatellites include CDS and UTR microsatellites based on an independent series.

microsatellites (148,464 out of 316,147), they are over-represented among EBML (**Fig 3**). On the other hand, coding microsatellites are under-represented (**Fig 3**). Enrichment analysis leading to these conclusions takes into account the coverage of microsatellites among samples and is robust to sample partitioning (see methods for details). Despite their under-representation, it is remarkable that coding EBML are found in all five ethnicities: 118 AFR, 51 EAS, 26 SAS, 20 EUR, and 14 AMR. Coding microsatellite in four genes (KAT6B, ATN1, HOMEZ, and CNDP1) are biased in all five ethnicities. Overall, we find 727 alleles for coding EBML. Two genes (ATN1 and VEZF1) have 13 alleles each: in both cases, the gene product harbors a glutamine repeat. Highly polymorphic– 10 alleles each–non-glutamine repeats are found in gene products for TRAK1 (glutamic acid), SUPT20HL1 (alanine), KDM6B (proline), and AUTS2 (histidine). On the contrary, TYW3 and MTERF4 only have two alleles; but, each has an allele only found in African samples. In the case of TYW3, 44 out of 654 African samples possess a 4-apartic acid (DDDD) allele. In the case of MTERF4, 39 out of 651 African samples possess a (glutamic acid/aspartic acid) 3-mer (DED).

## Characterization of EBML by repeat unit

In terms of repeat units, EBML are consistent with human microsatellites at large. Complete summary of EBML based on repeat unit is shown in **Fig 4A**. Mononucleotide poly-A/poly-T repeats are most common: we find 1,736 out of 3,984. These repeats are predominant in the genomes of human, *Drosophila melanogaster*, *Caenorhabditis elegans*, *Arabidopsis thaliana*, and *Saccharomyces cerevisiae* [39]. Mononucleotide poly-G/poly-C repeats are far less common in our findings (154 out of 3,984) and in the genomes of the aforementioned organisms; nevertheless, both classes are over-represented among EBML as are mononucleotide repeats overall (p = 2.2e-16) (**Fig 2C**). Investigation of dinucleotide motifs reveals that EBML AC repeats (735 total) are most common while GC repeats (5 total) are extremely rare; both of

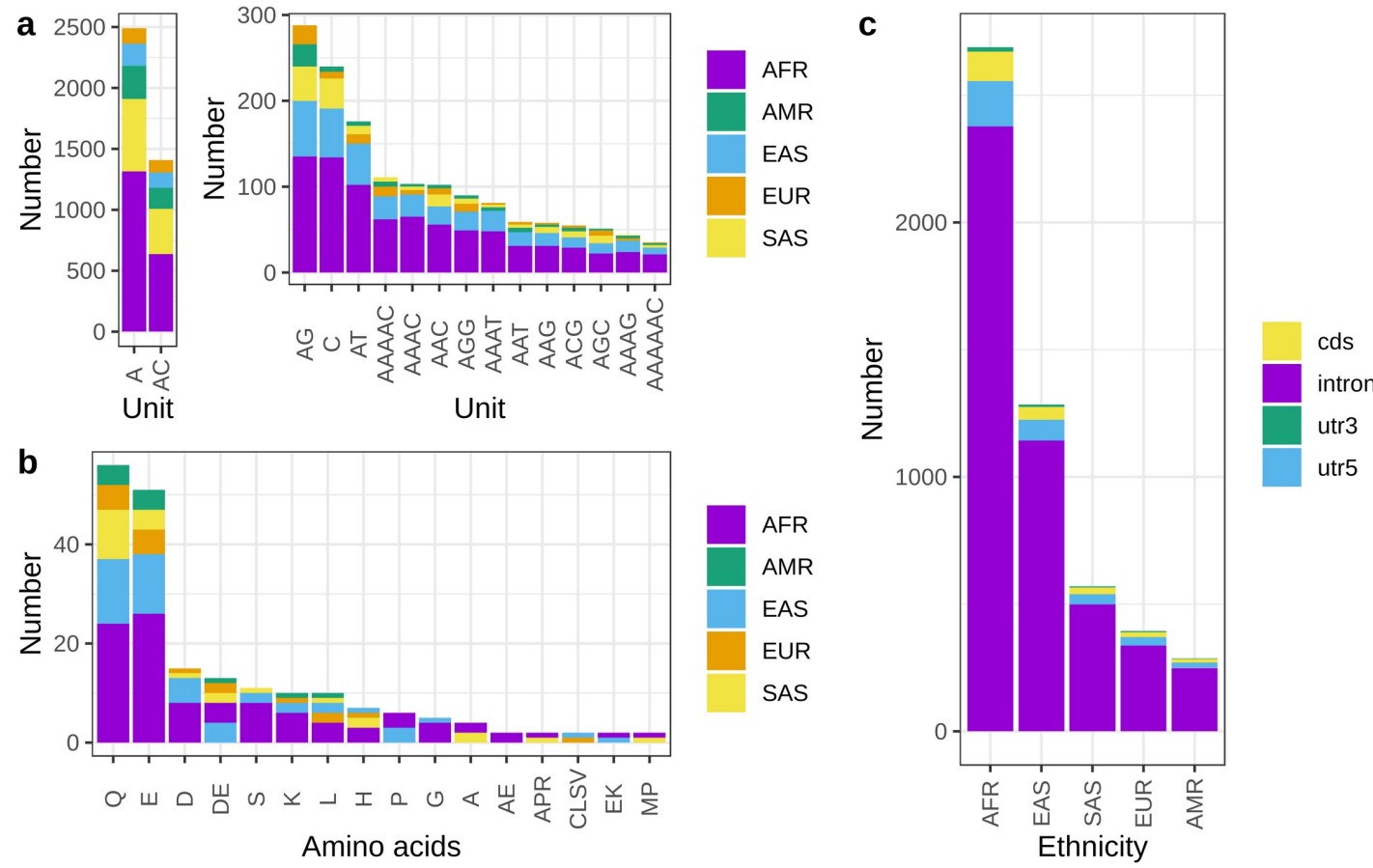

**Fig 4. Characterization of EBML by repeat unit, gene feature, and amino acid. a,** The 16 most common repeat motifs among all 3,984 EBML: colors indicate super-population (ethnicity); for clarity A/T and AC/TG/CA/GT repeats are shown separate. **b,** The 16 most common amino acid motifs for all 147 coding EBML: colors indicate super-population (ethnicity). Small and hydrophilic amino acids appear most frequently: glutamine (Q), glutamic acid (E), aspartic acid (D), serine (S), and lysine (K). Poly-glutamine repeats are over-represented (p = 9.32e-9) as are glutamic acid repeats (p = .0238) **c,** Summary of genomic regions harboring the 3,984 EBML in five super-populations (ethnicities). Colors indicate EBML overlap with gene introns, UTRs, and coding sequences. Note: exonic microsatellites (not shown) include CDS and UTR microsatellites.

these findings reiterate the overall distribution of genomic repeats in humans and D. melano-gaster [39]. EBML di-nucleotide repeats are over-represented (p = 2.23e-5) (**Fig 2C**). EBML tri-nucleotide repeats (356 total) are not over or under represented (p = .5812); however, EBML 4-mer, 5-mer, and 6-mer repeats are all under-represented (p<1e-15). Thus, the composition of EBML shifts from over-representation of mono-nucleotide and di-nucleotide repeats to under-representation of tetra-nucleotide repeats and beyond (**Fig 2C**).

## Poly-glutamine and poly-glutamic acid are common among coding EBML

It comes as no surprise that poly-glutamine repeats are most common among EBML found in coding sequences (we find 34 total); indeed, it is well known that glutamine repeats are abundant in the human proteome. Still, our results suggest that poly-glutamine is over-represented among coding EBML (p = 9.32e-9). Among the other common coding EBML–glutamic acid, aspartic acid, serine, and lysine–only the glutamic acid repeats (30 total) are over-represented (p = .0238). With the exception of serine, all of the aforementioned amino acids are hydrophilic. Overall we find hydrophilic amino acids (95 total) are over-represented (p = 2.38e-6) and hydrophobic amino acids (18 total) are under-represented (p = 1.20e-6) among the

EBML. Given these results it is remarkable that we find an EBML embedded in CNDP1 encoding an array of large hydrophobic leucine residues. Indeed it is thought that repeat expansions encoding hydrophobic amino acids are more likely to have deleterious effects on protein function [39]; yet, we find arrays of 4, 5, 6, and 7 residues. The protein coding microsatellite in ATN1 is remarkable for its degree of array length variation. We find 13 variants encoding a 6 to 17 residue glutamine repeat. Although these variants are well known and in the normal range [40, 41], our results show that the distribution of variants differs in the 5 super-populations (ethnicities) under investigation. A complete summary of the EBML protein coding repeats is shown in (**Fig 4B**).

## Pathway analysis for genes harboring EBML

Sets of genes harboring EBML for each ethnicity were checked for enrichment in Reactome and KEGG pathways [42, 43]; we find ten and nine significant pathways, respectively (**Fig 5**). The most significant Reactome pathways are those associated with the extracellular matrix (ECM): ECM proteoglycans, non-integrin membrane-ECM interactions, ECM organization, and degradation of the ECM. These four pathways–along with three collagen related pathways–were significant in two or more super-populations (**Fig 5A**). Statistically significant KEGG pathways reiterate these results. ECM receptor interaction and focal adhesions–specialized structures at cell-ECM contact points–are statistically significant in three and two super-populations, respectively (**Fig 5B**). Thus, the most significant functional enrichments–according to KEGG and Reactome–are those associated with the matrisome: a broad collection of ECM proteins comprising 1%-1.5% of the proteome. Further support for this conclusion is found by cross-referencing genes harboring EBML with 1,062 matrisome genes identified in a previous study [44]. Indeed core matrisome/matrisome associated genes are over-represented (p = .0023) (**Fig 2B; left panel**). In fact, 46 EBML are in collagen genes alone which in turn have been linked to numerous diseases (**S8 Table**). Still, the biological significance of these findings remains speculative. Only 64 out of 3,984 EBML have been shown to affect gene expression (see results above). Thus, more work needs to be done to determine the effects (if any) of EBML on ECM pathways and disease.

Despite our nascent understanding of repeat polymorphisms and gene expression, coding microsatellites are the established cause of numerous diseases [8]. Among the 147 coding EBML, we identify 14 with established links to diseases (**S10 Table**). Questions of prevalence or predisposition for these diseases among the super-populations we investigate could not be addressed. In particular, causative alleles for most repeat expansions diseases extend beyond the 100bp range of Illumina sequencing used for this study; this unfortunate limitation is further discussed below. Our data may shed light on susceptibility to knee and hip osteoarthritis stemming from an aspartic acid (D) repeat in the ASPN1 gene. Previous studies suggest that frequency of the D14 allele increases with disease severity [45, 46]. Our findings show that the D14 allele is most common in Africans which conceivably contributes to their high rates of large joint osteoarthritis [47]. Future work will address this hypothesis.

## Discussion

Microsatellites have been used for decades to study the genetic structure of diverse human populations [23–28]. Relatively recent studies have shed light on their role in complex trait heritability; epistatic interactions; and superiority as markers of genome integrity [48, 49]. Progress in these areas is important as microsatellites are thought to contribute to disease susceptibility and 'missing heritability' [7, 50]. Our work builds on what is known with four main results. First, we identify 3,984 EBML; genetic variation in these regions is representative of the

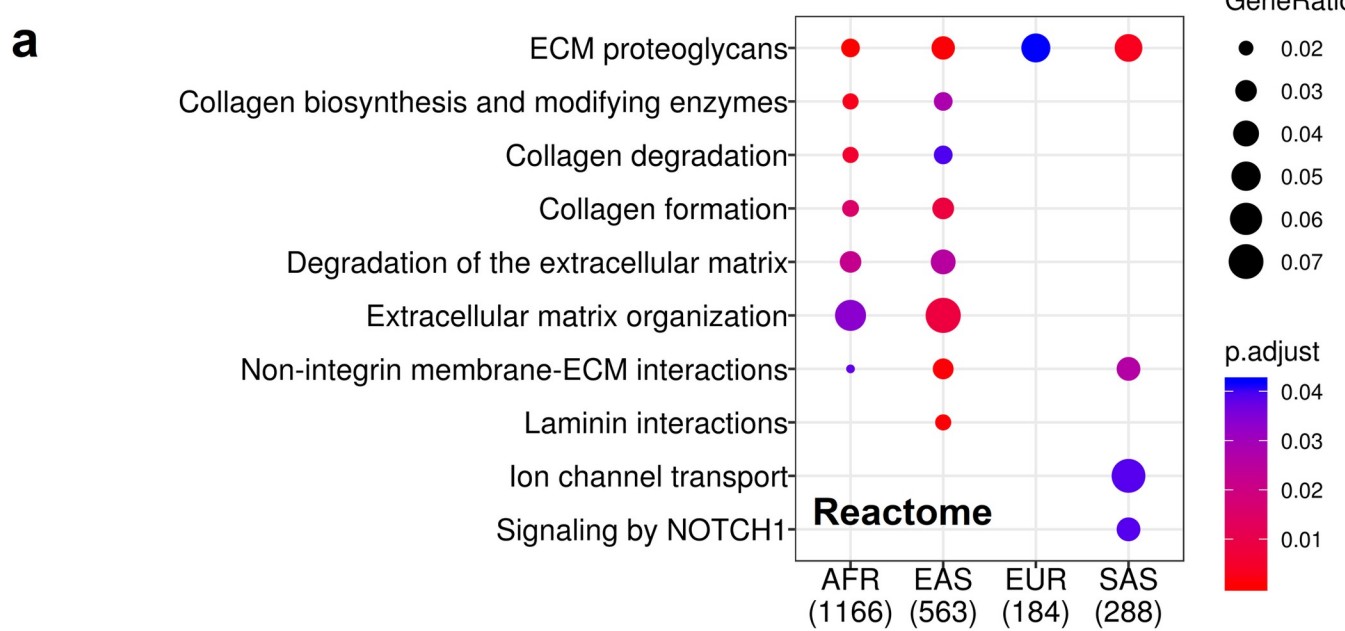

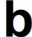

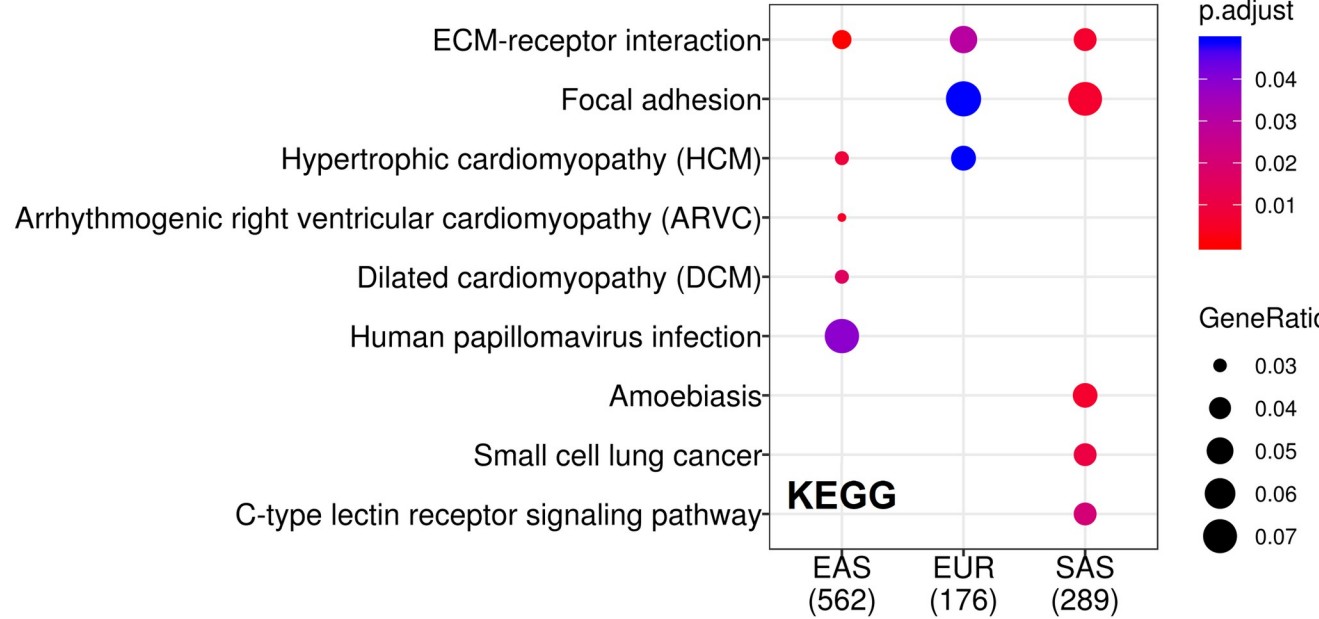

**Fig 5. Reactome and KEGG pathways over-represented by genes harboring EBML for each super-population. a,** The most significant Reactome pathways are those associated with the extra-cellular matrix (ECM) and collagen formation, degradation, and modification. **b,** The top two KEGG pathways–Focal adhesion and ECM-receptor interaction–are consistent with Reactome.

variation in 316,147 genome wide microsatellites. Second, a statistically significant number of EBML coincide with known eSTRs; i.e. they affect gene expression. Third, core matrisome and matrisome associated genes are over represented among genes harboring EBML. Fourth, a significant number of EBML are in putative selective sweep regions. The 3,984 EBML could be

useful in future studies of population structure and reiterate the utility of microsatellite analysis from Illumina sequencing [32].

Research leading to the discovery of eSTRs investigated their effects in different human populations; indeed, eSTR association signals were reproducible across population [9]. We build on these results by showing that some microsatellites likely contribute to differential gene expression in different populations. In fact, three of our main results support this overall conclusion: (a) EBML harbor genetic variation particular to one or more ethnicity; (b) EBML overlap eSTRs; and (c) EBML are over-represented in regions of soft selective sweep. On the other hand, we recognize that these results provide necessary but not sufficient indication that microsatellites have adaptive potential and are targets of positive selection. In particular, it is unclear if selection acting directly on microsatellites, with their enhanced mutability and extended sequence motifs, leave classical signatures of a selective sweep. However, we notice that, in a broad sense, a new microsatellite allele is an indel (insertion or deletion of (a) motif (s)), and indels or even transposable element insertions have been known to be targets of selection associated with selective sweeps [51, 52].

It is far too soon to say if functional repeats contribute to racial/ethnic disparities in complex disease rates and prevalence; if so, they are certainly one of many factors. Work on eSTRs demonstrated their enrichment in genes associated with various diseases: Crohn's disease, rheumatoid arthritis, and type 1 diabetes [9]. We find eSTRs over-represented among 3,984 EBML; however, other attempts to understand how microsatellites affect gene expression have led to mixed results. In particular, a study of nearly 5,000 promoter microsatellites revealed 183 significantly associated with nearby gene expression; but, only a small proportion of the microsatellites identified were significant in all populations under investigation [53]. Thus, more work needs to be done to establish links between eSTRs, EBML, and disease.

Some of our results are not surprising; in particular, characterization of the 3,984 EBML reiterates the extensive genetic diversity of African populations. Still, this is an important finding that underscores the lack of diversity in the human reference genome [54–56]. The current reference genome–GRCh38 –stems primarily from a single individual; consequently, routine genetic analysis such as read mapping and variant calling may suffer for individuals whose genetic makeup differs from the reference. Alternate loci and databases of known variants partially address this problem; however, many of the alleles we identify are not present in genomic databases such as dbSNP. In fact, the 3,984 EBML we identify have a combined 22,702 alleles rendering any single reference genome insufficient. Thus, our results reiterate the need to better represent diversity in the human genome and genomic databases at large [55]. Possibly, the EBML we identify–as well as undiscovered EBML–can help address this inequality by expanding what is known about human genetic variation.

Key limitations of our study lead us to hypothesize that many more EBML remain undiscovered. In particular, we only investigate microsatellites arrays shorter than 100bp: a number stemming from the short read (Illumina) sequencing used for the 1000 Genomes Project. Microsatellite variants exceeding 100bp are truncated rendering their true array length difficult to infer [57]. Advances in long read sequencing will help address this challenge but introduce others. Sequence alignment algorithms, especially those that use an affine gap model, can introduce errors when challenged with large insertions or deletions [58, 59]. Unfortunately, most clinically relevant microsatellite variants do exceed 100bp. In addition, the 1000 Genomes Project–and by extension our results–may still fall short in capturing genetic diversity in some populations. Principal component analysis has shown that genetic diversity in Southeast Asia is under-represented in the 1000 Genomes Project [60]; and, whole genome sequencing of African Populations has revealed millions of unshared genetic variants [56]. Thus, there are undoubtedly more EBML to be discovered.

## Methods

### Microsatellite list generation

A preliminary list of 625,178 microsatellites was generated using a two-step process: (a) detection in the reference genome, and (b) reduction of sequence similarity. Super-population specificity was subsequently tested for 316,147 microsatellites (see next sections). Here, we describe each step of preliminary list generation.

*(a) Detection of microsatellites in the reference genome*. A list of microsatellites in version 38 of the human reference genome was generated with a custom Perl script 'searchTandemRepeats.pl' using default parameters. This script has been used in previous microsatellite studies and is freely available online at http://genotan.sourceforge.net/#_Toc324410847 [61]. Briefly, the 'searchTandemRepeats.pl' script first searches for pure repetitive stretches: no impurities allowed. Imperfect repeats and compound repeats are handled using a "mergeGap" parameter with a default value of 10 base pairs. Essentially, impurities that interrupt stretches of pure repeat sequence are tolerated unless they exceed 10 base pairs. Likewise, repeats closer than 10 base pairs are considered compound. The initial list generated with this script included 1,671,121 microsatellites.

*(b) Reduction of sequence similarity*. It is well known that sequencing reads containing microsatellites are prone to mismapping [62]; particularly when different microsatellites possess the same repeat unit between similar 3' and 5' flanking sequences. We filtered the initial list of microsatellites (step a) to mitigate these effects. To begin, each microsatellite in the initial list (step a) was assigned a hash key constructed by concatenation of its 3' flanking sequence, repeat unit, and 5' flanking sequence. We used 5bp for the flanking regions. Thus, microsatellites 'GCTGC(A)$^{34}$CTTAG' and 'GCTGC(A)$^{15}$CTTAG' received the same hash key: 'GCTGCACT TAG'. Next, hash keys appearing more than once–and their corresponding microsatellites–were removed from the initial list. The fact that there were many of these potentially ambiguous regions is not surprising since microsatellites are often embedded in larger repetitive motifs such as LINES and SINES [63]. Our filtered preliminary list included 625,178 microsatellites unique in the human genome: available at http://www.cagmdb.org/view_micros.php

### Microsatellite genotyping

We used the program RepeatSeq [61] to determine the genotype of microsatellites in next generation sequencing reads. RepeatSeq operates on three input files: a reference genome, a file containing reads aligned to the human reference genome (.bam file), and a list of query microsatellites (see methods above). Like most microsatellite genotyping programs, RepeatSeq excludes reads that do not span the entire repeat. These unusable reads are detected by their lack of one or both flanking regions that anchor the microsatellite to a unique position in the genome; by default, RepeatSeq requires that each read contain at least 3 matching base pairs for both the 3' and 5' flanking region. To further increase the accuracy of genotyping calls, we only considered microsatellites with six or more mapped reads: by default genotype calls only require 2 reads. The advantage of RepeatSeq over other microsatellite genotyping programs is that it realigns each read to the reference genome prior to array length detection [57]. This mitigates the main cause of microsatellite genotyping errors, specifically, improper read alignments. Improper alignments often arise when the 3' and 5' flanking regions of a microsatellite mimic its repeat unit or from large insertions and deletions, which are common in microsatellites. In these situations, alignment algorithms often incorrectly open and extend gaps; thus, careful realignment is critical to accurate microsatellite array length detection. RepeatSeq has been used in previous studies of microsatellites and is freely available: https://github.com/adaptivegenome/repeatseq.

While other microsatellite genotypers report similar or better accuracy–hipSTR in particular [64]–RepeatSeq was specifically designed and validated using data from the 1000 Genomes Project. In addition, RepeatSeq preforms local realignment and multiple sequence alignment of microsatellite arrays. The lobSTR genotyping program–introduced in 2012 –does include analysis of the 1000 Genomes Project; however, evidence does not suggest that lobSTR assigns genotypes more accurately than RepeatSeq [65]. The exSTRa program is a new option which is more appropriate for detection of repeat expansion disorders [66]. Additional options include STRetch [67], TREDPARSE [68], and Dante [69].

## Samples

We used existing data from the 1000 Genomes Project. Specifically, samples were downloaded from phase 3 of the 1000 Genomes Project: ftp://ftp.1000genomes.ebi.ac.uk/vol1/ftp/phase3/. All samples listed in the phase 3 index– 20130502.phase3.exome.alignment.index–were included for analysis. Metadata for each sample was retrieved from the sample info file provided by the 1000 Genomes Project:

## Discovery and statistical testing of EBML

The 3,984 EBML presented in this work were discovered using a two step-procedure: (a) pairwise screening of super-populations, and (b) convergence of pairwise screens. In step (a) super-populations were screened pairwise: $\binom{5}{2} = 10$ total screens. The average number of tested microsatellites for each pairwise screen was 163,390 (**Fig 6A**). Step (b) only considered microsatellites passing false discovery: 3,713 on average (**Fig 6B**). In what follows, we describe steps (a) and (b) in detail.

*(a) Pairwise screening of super-populations*. Pairwise comparisons involved three sub-steps: *(i)* construct a *2xN* contingency table for each tested microsatellite; *(ii)* use Fisher's exact test to assign a p-value; and *(iii)* use the Benjamini-Hochburg multiple testing correction to mitigate false discovery (**Fig 6B**). Each *2xN* contingency table (sub-step *i*) was populated with the tally of samples in *2* super-populations over *N* different genotypes. At least 1 sample for each super-population was required; the number of genotypes (*N*) ranged from 1 (no variation) to over 20 (for highly polymorphic microsatellites). For each contingency table we performed a test of the null: no difference in genotype distribution. To assign a p-value (sub-step *ii*) we use Fisher's exact test. We considered using a $\chi^2$ test or G-test; however, Fisher's exact test has the advantage that it is equally valid on densely and sparsely populated tables. Various information metrics–which are not suited for testing a null–were not considered in this work. Only microsatellites passing false discovery (sub-step *iii*) were carried forward to step *(b)*, described below. The numbers of microsatellites in each pairwise screen and the numbers passing false discovery are shown in **Fig 6A**. Details for step (a) and sub-steps (i-iii) are shown in **Fig 6B**.

*(b) Convergence of pairwise screens*. Each super-population received 4 pairwise screens from step *(a)*; i.e. results of comparing microsatellite genotypes with the remaining 4 super-populations. Microsatellites for which the null was rejected in all 4 screens were considered EBML (**Fig 6B**). All intermediate calculations are provided in **S2 Dataset**.

## Annotation and enrichment analysis

Annotation and enrichment of the 3,984 EBML (see results) was performed with respect to (a) gene features, (b) known eSTRs, (c) matrisome genes, (d) selective sweep regions, (e) repeat lengths, (f) super-population pairs, and (g) amino acids. Analysis with respect to gene features (a) was designed to take into account differences in sample numbers: an intrinsic consequence

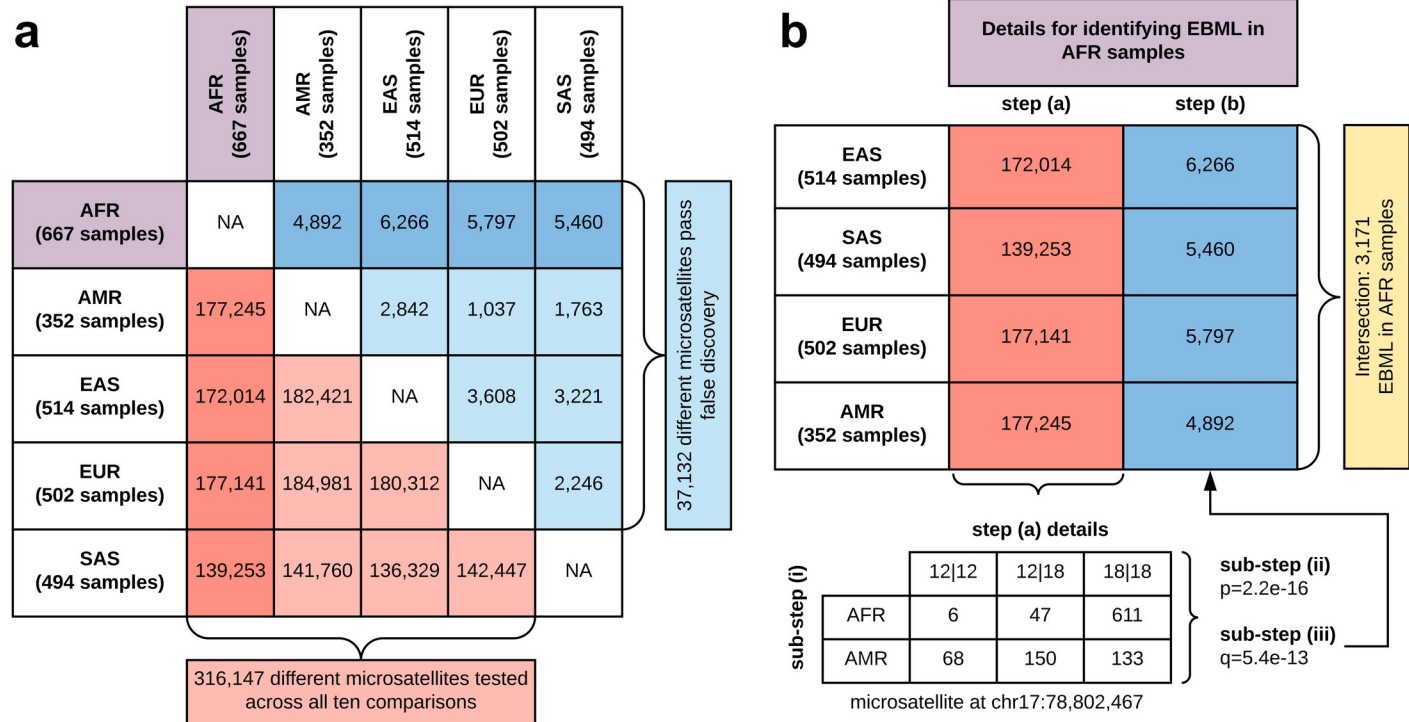

**Fig 6. Overall approach and statistical testing of genome wide microsatellites. a,** Number of microsatellites screened (red) and passing false discovery (blue) in pairwise comparisons of 5 super-populations. Details for one super-population (AFR) shown in bold. **b,** EBML for each super-population are identified using a two-step approach: step (a) pairwise screening against the remaining 4 super-populations, and step (b) convergence of the pairwise screens. Pairwise screens in step (a) involve 3 sub-steps: *(i)* construct a *2xN* contingency table for each microsatellite; *(ii)* Fisher's exact test to assign a p-value; *(iii)* Benjamini-Hochburg multiple testing correction to mitigate false discovery. Only microsatellites rejecting the null–no difference in genotype distribution–are examined in (b). EBML (yellow) are those that reject the null in all 4 pairwise screens: i.e. the set intersection of microsatellites identified in (b). We identify 3,171 EBML for AFR (shown), 450 for EUR, 1,494 for EAS, 647 for SAS, and 335 for AMR.

of the whole exome sequenced samples used for this study. Subsequent analyses (b-g) used 2x2 contingency tables with $\chi^2$ test of independence: details are described below.

*(a) Annotation and enrichment with respect to gene regions.* We used cruzdb–a freely available package for Python–to identify microsatellite overlaps with known introns, exons, coding sequences (cds), 3'-UTRs, and 5'-UTRs. Which (if any) of these features are enriched among the EBML? To check for enrichments we constructed a 2x2 contingency table for each region based on classifications of microsatellites: EBML; not EBML; overlap with gene region; no overlap with gene region. We performed a $\chi^2$ test of the null: no association between EBML and the gene region (intron, exon, 3'-UTR, 5'-UTR, or cds). However, a single $\chi^2$ test of independence was insufficient since the samples used for microsatellite genotyping (see above) were whole exome sequenced. Consequently, coding and exonic microsatellites tended to be present in more samples than other gene features; on the contrary, intron and UTR microsatellites tended to be present in fewer samples. Thus, an intrinsic difference in the composition of tested microsatellites and EBML was anticipated.

We therefore preformed a series of enrichment tests beginning with the aforementioned $\chi^2$ test (this is the first iteration of the series). Successive iterations of the series removed microsatellites from both lists (all tested and EBML) and recomputed enrichment p-values for introns, exons, cds, 3'-UTRs, and 5'-UTRs. Removal of microsatellites in each iteration was determined by sample size: the second iteration removed microsatellites sequenced in 4 or fewer samples;

the third iteration removed microsatellites sequenced in 8 or fewer samples. Minimum sample size doubled for 10 iterations.

A second series of tests used a variation of this approach to demonstrate robustness of enrichment analysis to procedure details. Again, each iteration of the series recomputed enrichment p-values for introns, exons, cds, 3'-UTRs, and 5'-UTRs. The first iteration considered microsatellites possessed by 2–3 samples. The second iteration considered microsatellites possessed by 4–7 samples. Window size doubled for 10 iterations. Conclusions drawn from enrichment analysis (see results) were the same regardless of approach. Both approaches suggest that intronic microsatellites are over-represented, and coding microsatellites are under-represented among those that are super-population specific.

*(b) Annotation and enrichment with respect to significant expression simple tandem repeats (eSTRs)*. A previous survey of 80,980 short tandem repeats (STRs) identified 2,060 that contribute to gene expression in human (eSTRs) [9]: 13,259 (of the 80,980) are present in our survey. We constructed a 2x2 contingency table based on classifications of the 13,259 shared repeats: EBML; not EBML; affects gene expression (eSTR); does not affect gene expression (other STR). We performed a $\chi^2$ test of the null: no association between EBML and eSTRs. The null was rejected (p = 1.53e-8); see results.

*(c) Annotation and enrichment with respect to matrisome genes*. A previous study identified 1,062 core matrisome and matrisome associated genes [44]. We constructed a 2x2 contingency table based on classifications of microsatellites: EBML; not EBML; core matrisome/matrisome associated gene; any other gene. We performed a $\chi^2$ test of the null: no association between EBML and matrisome genes. The null was rejected (p = .0023); see results.

*(d) Annotation and enrichment with respect to selective sweep regions*. Putative selective sweep regions were previously identified in 6 populations of the 1000 Genomes Project [34]: Three African populations (YRI, GWD, and LWK); one European population (CEU); one East Asian population (JPT); and one American population (PEL). A 2x2 contingency table was constructed based on classifications of microsatellites: EBML; not EBML; within sweep region; outside of sweep region. We performed a $\chi^2$ test of the null: no association between EBML and sweep regions. The null was rejected ($p = 0.018$); see results.

*(e) Enrichment with respect to repeat length*. Six 2x2 contingency tables were constructed based on classifications of microsatellites by super-population specificity and unit length. In each case we perform a $\chi^2$ test of the null: no association between EBML and unit length (see results).

*(f) Enrichment with respect to specificity in two or more super-populations*. Ten 2x2 contingency tables were constructed for the 3,984 EBML (one table for each pair of super-populations). Table entries enumerated the number of EBML in both super-populations, one super-population, and neither super-population, respectively. We performed a $\chi^2$ test of the null: no association between EBML in the pair of super-populations; see results.

*(g) Enrichment with respect to amino acids for coding microsatellites*. Contingency tables were constructed for coding microsatellites based on classification by super-population specificity and amino acids. We performed a $\chi^2$ test of the null: no association between EBML and amino acids; see results.

## Principal component analysis

Smartpca with default parameters from EIGENSOFT [70] was used to perform principal component analysis (PCA) based on a covariance matrix. Details of EIGENSOFT can be found elsewhere. Briefly, analysis began with a matrix ($X_{mn}$) populated with microsatellite genotypes ($m$ = 3,984) for each individual ($n$ = 2,529). For PCA analysis of all tested microsatellites we

used $m$ = 316,167. Each matrix row ($m$) received two types of normalization. First, genotype (row) means are set to zero by subtracting the quantity $\mu_m = (\Sigma_n X_{mn})/N$ from each entry. Second, rows were normalized by dividing each entry by $\sqrt{(p_m(1 - p_m))}$: see [70] for details. Once normalized, the covariance was computed for each pair ($n \cdot n$) of individuals; the result is a covariance matrix ($C_{n \; by \; n}$). Eigenvectors of the covariance matrix (typically ranked by their eigenvalue) represent the principal components of variation: see [70] for full details.

## Pathway analysis

Analysis of metabolic pathways harboring EBML was performed using the Reactome database and the Kyoto Encyclopedia of Genes and Genomes (KEGG). To begin, we used cruzDB to identify genes harboring the 3,984 EBML: 3,171 for AFR, 450 for EUR, 1,494 for EAS, 647 for SAS, and 335 for AMR. CruzDB did not identify a gene for 200 EBML. Next, the gene lists for each super-population were submitted to Reactome and KEGG databases: submission was performed programmatically using the clusterProfiler package in R [71]. The function compareCluster was used for visualization.

## Sample availability, code availability, and URLs

Existing data was used for this study. All samples are freely available from the 1000 Genomes Project: http://www.internationalgenome.org/. PCA was performed with EIGENSOFT: https://github.com/DReichLab/EIG. Fourfold plots for matrisome and eSTR enrichments use the visualizing categorical data (vcd) package available in R. UpSet plot and 5-way Venn diagram use the UpSetR and Venn packages available in R, respectively. Kegg and Reactome pathway analysis was performed with the ReactomePA package and visualized with the clusterProfiler package. Additional R packages included corrplot, questionr, and analysis of biological data (abd). Sequencing reads, alignments, and microsatellite genotypes are available online: www.cagmdb.org/.

## Supporting information

**S1 Table. List of 3,984 EBML identified in this study.**
(XLSX)

**S2 Table. List of 3,171 microsatellites specific to African populations.**
(XLSX)

**S3 Table. List of 1,494 microsatellites specific to East Asian populations.**
(XLSX)

**S4 Table. List of 647 microsatellites specific to South Asian populations.**
(XLSX)

**S5 Table. List of 450 microsatellites specific to European populations.**
(XLSX)

**S6 Table. List of 335 microsatellites specific to American populations.**
(XLSX)

**S7 Table. List of 64 EBML also identified as eSTRs.**
(XLSX)

**S8 Table. List of 232 EBML in matrisome core/associated genes.**
(XLSX)

**S9 Table. List of 21 EBML coding repeats in selective sweep regions.**
(XLSX)

**S10 Table. List of 14 EBML coding repeats previously implicated in disease.**
(XLSX)

**S1 Dataset. Genotyping data for 3,984 EBML in 1000 Genomes Project samples.**
(BZ2)

**S2 Dataset. Pairwise comparisons of microsatellites in 5 super-populations.**
(7Z)

**S1 Code. Functions to access genotype tables for EBML.**
(R)

## Acknowledgments

NAK, LK, RA, PM, and HRG contributed to the conceptualization of this project, experimental design, and data analysis. NAK, LK, and PM contributed the writing of this manuscript. NAK, LE, PS, LK, and PM were responsible for software writing and data analysis. NAK, AP, LE, LK, RA, PM, HRG, and RTV were responsible for manuscript preparation. All authors read and approved the final manuscript. We thank Liang Shan for assisting with statistical analysis.

## Author Contributions

**Conceptualization:** Nick Kinney, Lin Kang, Ramu Anandakrishnan, Robin T. Varghese, P. Michalak, Harold R. Garner.

**Data curation:** Nick Kinney, Laurel Eckstrand, Arichanah Pulenthiran, Peter Samuel, Ramu Anandakrishnan, Robin T. Varghese, P. Michalak, Harold R. Garner.

**Formal analysis:** Nick Kinney, Lin Kang, Ramu Anandakrishnan, P. Michalak, Harold R. Garner.

**Funding acquisition:** Nick Kinney, Harold R. Garner.

**Investigation:** Nick Kinney, Lin Kang, Laurel Eckstrand, Arichanah Pulenthiran, Peter Samuel, Ramu Anandakrishnan, Robin T. Varghese, P. Michalak, Harold R. Garner.

**Methodology:** Nick Kinney, Lin Kang, Ramu Anandakrishnan, P. Michalak, Harold R. Garner.

**Project administration:** Nick Kinney, Harold R. Garner.

**Resources:** Nick Kinney, Harold R. Garner.

**Software:** Nick Kinney, Lin Kang, Ramu Anandakrishnan, P. Michalak.

**Supervision:** Harold R. Garner.

**Validation:** Nick Kinney, Laurel Eckstrand.

**Visualization:** Nick Kinney, Lin Kang.

**Writing – original draft:** Nick Kinney, Lin Kang, P. Michalak, Harold R. Garner.

**Writing – review & editing:** Nick Kinney, Laurel Eckstrand, Arichanah Pulenthiran, Harold R. Garner.

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
