## [Decision Letter · Decision Letter 0]

19 Sep 2019

PONE-D-19-21466

Abundance of Super-Population Specific Microsatellites in Human Gene Regions

PLOS ONE

Dear Dr. Kinney,

Thank you for submitting your manuscript to PLOS ONE. After careful consideration, we feel that it has merit but does not fully meet PLOS ONE’s publication criteria as it currently stands. Therefore, we invite you to submit a revised version of the manuscript that addresses the points raised during the review process.

The reviewers highlight multiple points that need consideration.

This involves framing of the study (what are the central questions), definitions of terms (super or just specific), methodogical concerns (exome sequencing origin?) and questions of interpretation.

We would appreciate receiving your revised manuscript by Nov 03 2019 11:59PM. To enhance the reproducibility of your results, we recommend that if applicable you deposit your laboratory protocols in protocols.io, where a protocol can be assigned its own identifier (DOI) such that it can be cited independently in the future. For instructions see: http://journals.plos.org/plosone/s/submission-guidelines#loc-laboratory-protocols

We look forward to receiving your revised manuscript.

Kind regards,

Arnar Palsson, Ph.D.

Academic Editor

PLOS ONE

Journal Requirements:

2. Please amend either the author names on the online submission form (via Edit Submission) or in the manuscript so that they are identical.

Additional Editor Comments (if provided):

The reviewers highlight multiple points that need consideration.

This involves framing of the study (what are the central questions), definitions of terms (super or just specific), methodogical concerns (exome sequencing origin?) and questions of interpretation.

Reviewers' comments:

Reviewer's Responses to Questions

**Comments to the Author**

1. Is the manuscript technically sound, and do the data support the conclusions?

Reviewer #1: Partly

Reviewer #2: Partly

2. Has the statistical analysis been performed appropriately and rigorously? 

Reviewer #1: Yes

Reviewer #2: No

3. Have the authors made all data underlying the findings in their manuscript fully available?

Reviewer #1: Yes

Reviewer #2: Yes

4. Is the manuscript presented in an intelligible fashion and written in standard English?

Reviewer #1: Yes

Reviewer #2: Yes

5. Review Comments to the Author

Reviewer #1: Kinney et al. investigate the variation of more than 300,000 microsatellites across five super-populations of human ethnic groups using genotypes called from 100bp Illumina reads as part of the 1000 Genomes Project. The authors identify 3,984 microsatellites they term super-population specific (SPS). Subsequent analyses of SPS microsatellites reveal: (1) PCA on the original set of 300,000+ microsatellites and the reduced set of SPS microsatellites produce highly similar patterns on PC1-PC2 biplots; (2) a significant fraction of SPS microsatellites overlap with a set of previously identified eSTRs – microsatellites associated with differences in gene expression; (3) genes harboring SPS microsatellites are enriched for extra-cellular matrix pathways, and; (4) a significant number of SPS microsatellites are coincidental with sequences previously implicated in selective sweeps.

Major Concerns

1. The motivation for identifying SPS microsatellites is never explicitly stated. The authors simply tell us they have done so. The closest we get to a motivating statement is in the abstract: “We discover 3,984 super-population specific microsatellites and investigate their potential biological and clinical significance.” But the connection between SPS microsatellites and their functional significance seems hinted at rather than explicitly reasoned through. We’re told, for example, that the SPS set is enriched for eSTRs under a section heading of “Numerous SPS microsatellites affect gene expression.” Yet, we never get explicit statements relating to the importance of this finding. We are told there is a significant overlap, that it may or may not relate to ethnic difference in disease prevalence, and that is it. While I applaud the Introduction’s excellent summary of the state of research into functional microsatellites, the reader is left to connect dots that are too far apart.

The four main findings emphasized by the authors – particularly (2) and (4) above – suggest the main motivation for identifying SPS microsatellites is that differences in genotype frequencies among super populations are potentially explained by the action of natural selection … and are therefore of functional importance. However, the manner in which the paper is written makes it feel like we should just know why SPS microsatellites are of importance and that we should just know why the statistically significant findings of the authors prove the worth of SPS microsatellites.

2. The definition of a super-population specific microsatellite is never given. While the method for finding them is explained well and in detail in the Methods section, a succinct, intuitive definition should be given in the abstract, as well as the Introduction and/or Results. For example, the authors might write, “An SPS microsatellite is a microsatellite in which the genotype frequencies of at least one super-population differs from those of all four of the remaining super-populations.” This is still a little clunky, but something to that effect would have made my reading of the paper much easier from the beginning.

3. Although differences in genotype frequencies may in fact stem from the action of natural selection due to different selective pressures at the continental scale, the authors never mention alternative explanations. An obvious alternative cause of SPS patterns of microsatellite variation is the combination of high mutability, genetic drift, and isolation-by-distance. These population genetic factors could easily lead to substantial differences in populations separated by vast distances. Indeed, the main interest in microsatellites has long been their potential to diagnose population structure. The authors do mention the potential use of microsatellites for population structure analyses near the beginning of the Discussion. However, the Discussion should include an honest appraisal of alternative evolutionary explanations for the development of SPS loci.

4. Another example of hinting at the importance of the findings rather than appraising them in a comprehensive and objective manner is the overlap of SPS microsatellites with genomic regions previously identified as potential regions where natural selection acted. The authors write that a selective sweep is “due to a novel allele [that] reduces nearby genetic variation.” However, natural selection on a microsatellite is a very different animal than selection on a SNP.. Selective sweeps of linked genetic variation result from the emergence of that novel allele on a specific genetic background. If selection actually acts on a microsatellite locus, a novel allele is difficult to define. Due to the high mutability of microsatellite loci, the same favored repeat length can arise many times, linked to many different combinations of linked SNPs. Therefore, it is not clear that selection on a microsatellite locus would lead to the classical patterns associated with selective sweeps. The authors do mention that the overlaps are with soft sweep regions, which themselves leave much messier signatures of natural selection than hard sweeps on a new variant. In my opinion, however, this still overlooks the inherent problem of attempting to link selection on microsatellites with signatures of selection based on theory that assumes a SNP is the target of selection.

Indeed, we might think of this in the reverse. Isn’t it possible that selection on a single nucleotide variant (that may only offer a selective advantage in only one or a subset of super populations) would cause patterns of variation at linked microsatellites to become different from each other in different populations? In other words, the differences in microsatellite variation point to selection, but selection on a SNP. This is exactly how selection on SNPs in cis-regulatory regions that lead to lactase persistence in some African populations was identified.

Minor Concerns

1. p. 2, ln 44: The authors compare array length variation, not “array length mutations”

2. (throughout) Principal component analysis, not Principle component analysis.

3. (throughout) Strange use of colons. For the most part, the authors seem to use them like an em dash would be used.

4. p. 6, ln 116: just curious if higher PCs (e.g., PC3 or PC4) lead to separation of AMR and SAS super populations. More importantly, how much variation do the first two PCs explain?

5. p. 7, lns 143-146. The authors state that eSTRs are commonly found in regions subject to purifying selection, then define a sweep in terms of positive selection.

6. p. 8, ln 157+ Confused by the genomic compartments used. UTRs are technically part of exons, but not the CDS. So, (1) you might make a distinction between UTRs and full+partial exons that make up the CDS … but then why a separate “coding” compartment? … or (2) you might make a distinction between UTRs and CDS … but then why a separate “exon” compartment, since UTRs and CDS comprise exons. Is there overlap between the compartments here? – i.e., are some STRs double-counted?

7. p 11., ln 197 Reference to support the claim that hydrophobic amino acids are more likely to have deleterious effects on protein function.

8. p 14., 1st paragraph Why devote a whole paragraph of a short discussion section to the idea that we have an incomplete picture of human genetic variation. Isn’t the 1000 Genomes Project (with its 26 individual populations) a big step in the right direction? These days, the reference genome is simply a jumping off point for analyses that do make use of data sets that include much more comprehensive coverage of human genetic variation.

Reviewer #2: The manuscript is interesting but several issues need to be addressed before publication in PLOSONE.

1. What were the default parameters of the perl script the authors used for finding the repeats? Did they use a constant length cutoff, or did it change with the repeat unit? Studies have reported way more than 1.6 million STRs in humans, and knowing the exact parameters the authors used will help readers understand why they had such a small set of loci to start with.

2. Line 112 and other relevant places should be modified to clearly convey that they have used existing data and not done the sequencing themselves.

3. Link to eSTRs - Why did the authors choose to use all 80,980 loci rather than the 2060 loci that showed a significant effect on expression as reported by Gymrek et al. ? How many common loci are there between the 2060 eSTRs and 3984 SPS?

4. Authors are advised to elaborate on the p value testing they did to show that 64 common out of 3984 total is significant. In particular, I would like to see if they took into account the bias towards coding regions that is introduced by whole exome sequencing. As can be seen from their study, the loci considered by Gymrek et al were predominantly proximal to genic regions. And as the current study preliminarily uses exome data, the chances of overlap are higher than what is true if whole genome is considered as background. The authors must incorporate this into calculating the significance.

5. Regarding the analysis of over/under representation by repeat unit length, if a larger length cutoff was used for larger repeat unit lengths, that might explain why tetra to hexamers are underrepresented in their analyses.

6. The authors call these microsatellites super-population specific, yet there are a lot of overlaps of these loci indicating they are merely highly polymorphic. Is it really correct to call them “specific”? On a related note, what happens to the analyses if they are done only with the unique loci (~2500 loci added together).

6. PLOS authors have the option to publish the peer review history of their article (what does this mean?). If published, this will include your full peer review and any attached files.

Reviewer #1: No

Reviewer #2: No

---

## [Author Response · Author response to Decision Letter 0]

3 Oct 2019

(images included - see the attached document)

Dr. Arnar Palsson, Ph.D.

Academic Editor of Plos One

Dear Dr. Palsson:

We are pleased that our manuscript entitled “Abundance of Ethnically Biased Microsatellites in Human Gene Regions” is accepted for publication in Plos One. In the pages that follow we respond point-by-point to each of the reviewer concerns. The title of the original manuscript was “Abundance of Super-Population Specific Microsatellites in Human Gene Regions”. This change stems from reviewer concerns regarding definition of terms. It is more accurate to call these regions bias rather than specific; and, ethnicity has been used previously as a surrogate for the less familiar term super-population. We have addressed a number of additional weaknesses and look forward to sharing our revised manuscript with the scientific community. There are no related articles published or under consideration. If there are any specific questions that need a conversation, my cell is 301 338-1181.

Sincerely,

Dr. Nicholas Kinney

Primary Care Research Network and the Center for Bioinformatics and Genetics, VCOM

Professor of Biomedicine, VCOM

Reviewer #1: 

Kinney et al. investigate the variation of more than 300,000 microsatellites across five super-populations of human ethnic groups using genotypes called from 100bp Illumina reads as part of the 1000 Genomes Project. The authors identify 3,984 microsatellites they term super-population specific (SPS). Subsequent analyses of SPS microsatellites reveal: (1) PCA on the original set of 300,000+ microsatellites and the reduced set of SPS microsatellites produce highly similar patterns on PC1-PC2 biplots; (2) a significant fraction of SPS microsatellites overlap with a set of previously identified eSTRs – microsatellites associated with differences in gene expression; (3) genes harboring SPS microsatellites are enriched for extra-cellular matrix pathways, and; (4) a significant number of SPS microsatellites are coincidental with sequences previously implicated in selective sweeps.

Major Concerns

1a. The motivation for identifying SPS microsatellites is never explicitly stated. The authors simply tell us they have done so. The closest we get to a motivating statement is in the abstract: “We discover 3,984 super-population specific microsatellites and investigate their potential biological and clinical significance.” But the connection between SPS microsatellites and their functional significance seems hinted at rather than explicitly reasoned through. 

We agree that the motivation is tacitly assumed when it should be explicitly stated in the abstract; the reviewer’s thorough suggestions have enabled us to address this weakness. The revised manuscript now follows a clear thread from the beginning of the abstract which begins by highlighting the emerging functional relevance of some microsatellites: 

"[Microsatellites] have been used for decades as putatively neutral markers to study the genetic structure of diverse human populations. However, recent studies have demonstrated that that some microsatellites contribute to gene expression, cis heritability, and phenotype."

Next, we emphasize the motivation for the current study and approach:

"As a corollary, some microsatellites may contribute to differential gene expression and RNA/protein structure stability in distinct human populations. To test this hypothesis we investigate genotype frequencies, functional relevance, and adaptive potential of microsatellites in five super-populations (ethnicities) drawn from the 1000 Genomes Project."

Although we have addressed the reviewers comment economically, we think the improvement is significant.

1b. We’re told, for example, that the SPS set is enriched for eSTRs under a section heading of “Numerous SPS microsatellites affect gene expression.” Yet, we never get explicit statements relating to the importance of this finding. We are told there is a significant overlap, that it may or may not relate to ethnic difference in disease prevalence, and that is it. While I applaud the Introduction’s excellent summary of the state of research into functional microsatellites, the reader is left to connect dots that are too far apart.

We agree that this is an important section that was given a weak heading and interpretation. The new section heading helps the reader anticipate the importance of the overlap between SPS microsatellites (now referred to as EBML), eSTRs, and selective sweeps:

"Correspondence between EBML, eSTRs, and selective sweeps suggest adaptive potential"

We have added a paragraph at the end of this important section that helps the reader interpret the correspondence between EBML, eSTRs, and selective sweeps. We begin by reiterating the definition of each region and provide a clear statement of significance.

"Overall, these findings suggest a degree of mutual overlap between EBML, eSTRs, and selective sweeps in the human genome. The interpretation of this overlap is built up from the definition of each region. Briefly, each eSTR has the capacity to affect gene expression; and, for each EBML at least one ethnicity has a distribution of genotypes statistically different from the remaining four. Thus, any correspondence between the two implies differential gene expression in one or more human super-populations."

We build upon this interpretation by offering alternative explanations based on high mutability, genetic drift, isolation-by-distance, and natural selection. The finding that EBML overlap with selective sweeps now has a logical motivation; in particular, it is a necessary way to show that microsatellites have adaptive potential:

"These differences likely stem from a complex combination of high mutability, genetic drift, isolation-by-distance, and natural selection. Drift, mutability, and isolation are inevitable; but, the situation is less clear for natural selection. The correspondence between EBML and selective sweeps suggests they have adaptive potential and may be targeted by natural selection"

Alternative explanations are provides and we offer a measured overall conclusion that reiterates the section heading:

"On the other hand, it’s possible that EBML are simply in linkage disequilibrium with targets of selection such as nearby SNPs. Overall it seems likely that some EBML – particularly the ones that overlap eSTRs – have bona fide adaptive potential."

1c. The four main findings emphasized by the authors – particularly (2) and (4) above – suggest the main motivation for identifying SPS microsatellites is that differences in genotype frequencies among super populations are potentially explained by the action of natural selection … and are therefore of functional importance. However, the manner in which the paper is written makes it feel like we should just know why SPS microsatellites are of importance and that we should just know why the statistically significant findings of the authors prove the worth of SPS microsatellites.

We agree that (2) and (4) are highly non-trivial, complementary, and represent the main findings of our work. This was not made clear enough to the reader. We have added passages throughout the manuscript to clarify the argument and recognize alternative explanations. The first section of our results now includes the following interpretation:

"Identifying EBML is an easier task than explaining their origin. EBML could emerge due to high microsatellite mutability, genetic drift, isolation-by-distance, or natural selection. The action of natural selection – investigated in the next section – would suggest adaptive potential."

Have identified SPS microsatellites (now referred to as EBML) in the first section; this should que the reader that the manuscript will be turning to questions of their significance. The second section of the results builds a case for adaptive potential and suggests alternative explanations (see the new interpretation paragraph on page 9 lines 179-190):

"Overall it seems likely that some EBML – particularly the ones that overlap eSTRs – have bona fide adaptive potential."

2. The definition of a super-population specific microsatellite is never given. While the method for finding them is explained well and in detail in the Methods section, a succinct, intuitive definition should be given in the abstract, as well as the Introduction and/or Results. For example, the authors might write, “An SPS microsatellite is a microsatellite in which the genotype frequencies of at least one super-population differs from those of all four of the remaining super-populations.” This is still a little clunky, but something to that effect would have made my reading of the paper much easier from the beginning.

We have formulated a one-sentence minimal definition and now refer to these regions as ethnically biased microsatellite loci (EBML). The abstract now states the definition of these regions:

"We discover 3,984 ethnically biased microsatellite loci (EBML); for each EBML at least one ethnicity has genotype frequencies statistically different from the remaining four."

The same definition is provided in the author summary, introduction, and results. Changing the name of these regions is motivated by a need for clarity. As pointed out by reviewer 2 it is not appropriate to call these regions “specific”. Indeed the underlying genotype distributions may overlap and the same region may be significant in more than one super-population; calling these regions bias is more accurate. Ethnicity has been used previously as a surrogate for the less familiar term super-population.

3. Although differences in genotype frequencies may in fact stem from the action of natural selection due to different selective pressures at the continental scale, the authors never mention alternative explanations. An obvious alternative cause of SPS patterns of microsatellite variation is the combination of high mutability, genetic drift, and isolation-by-distance. These population genetic factors could easily lead to substantial differences in populations separated by vast distances. Indeed, the main interest in microsatellites has long been their potential to diagnose population structure. The authors do mention the potential use of microsatellites for population structure analyses near the beginning of the Discussion. However, the Discussion should include an honest appraisal of alternative evolutionary explanations for the development of SPS loci.

Agreed, we have made a substantial revision to the discussion to address this weakness. Specifically, we have added a new paragraph that addresses the fundamental mechanisms of population differentiation (high microsatellite mutability, genetic drift, isolation-by-distance, and natural selection). We reiterate which of our results suggest some microsatellites have adaptive potential while recognizing alternatives:

"these results provide necessary but not sufficient indication that microsatellites have adaptive potential and are targets of positive selection. In particular, it is unclear if selection acting directly on microsatellites, with their enhanced mutability and extended sequence motifs, leave classical signatures of a selective sweep. However, we notice that, in a broad sense, a new microsatellite allele is an indel (insertion or deletion of (a) motif(s)), and indels or even transposable element insertions have been known to be targets of selection associated with selective sweeps (51,52)"

This paragraph (in addition to the new passages in the results) should clarify the message of the paper and offer further lines of inquiry to researches in the field. In addition, we have removed a discussion paragraph regarding codon and amino acid usage in lower eukaryotes. While interesting, this passage detracts from the focus of the paper which is human genetics.

4. Another example of hinting at the importance of the findings rather than appraising them in a comprehensive and objective manner is the overlap of SPS microsatellites with genomic regions previously identified as potential regions where natural selection acted. The authors write that a selective sweep is “due to a novel allele [that] reduces nearby genetic variation.” However, natural selection on a microsatellite is a very different animal than selection on a SNP.. Selective sweeps of linked genetic variation result from the emergence of that novel allele on a specific genetic background. If selection actually acts on a microsatellite locus, a novel allele is difficult to define. Due to the high mutability of microsatellite loci, the same favored repeat length can arise many times, linked to many different combinations of linked SNPs. Therefore, it is not clear that selection on a microsatellite locus would lead to the classical patterns associated with selective sweeps. The authors do mention that the overlaps are with soft sweep regions, which themselves leave much messier signatures of natural selection than hard sweeps on a new variant. In my opinion, however, this still overlooks the inherent problem of attempting to link selection on microsatellites with signatures of selection based on theory that assumes a SNP is the target of selection.

Indeed, we might think of this in the reverse. Isn’t it possible that selection on a single nucleotide variant (that may only offer a selective advantage in only one or a subset of super populations) would cause patterns of variation at linked microsatellites to become different from each other in different populations? In other words, the differences in microsatellite variation point to selection, but selection on a SNP. This is exactly how selection on SNPs in cis-regulatory regions that lead to lactase persistence in some African populations was identified.

The reviewer is certainly right: the association between microsatellite alleles and selective sweep regions does not necessarily imply that a microsatellite allele drives selective sweep. The alternative scenario, whereby a neighboring SNP drives selective sweep, while the microsatellite hitchhikes to high frequency as a result of its linkage to the SNP, is equally, if not more, likely, and we now clarify it in the manuscript:

"…it’s possible that EBML are simply in linkage disequilibrium with targets of selection such as nearby SNPs. Overall it seems likely that some EBML – particularly the ones that overlap eSTRs – have bona fide adaptive potential."

The reviewer also brings up an important issue whether or not microsatellites are expected to produce classical selective sweep patterns. We believe that as long as a new microsatellite allele increases fitness, positive selection acting on it will leave a signature similar to a SNP-based sweep. In a broader sense, at the sequence polymorphism level, a new microsatellite allele is an indel (insertion of deletion), and indels or even transposable element insertions have been known to be targets of selection associated with selective sweeps (PMID: 20333210, 14745026). Higher mutation rate of microsatellites may actually increase the contribution from hard selective sweeps that typically originate from newly emerging mutations, relative to soft sweeps that leverage the pre-existing standing genetic variation. We clarify this in Discussion as follows:

"On the other hand, we recognize that these results provide necessary but not sufficient indication that microsatellites have adaptive potential and are targets of positive selection. In particular, it is unclear if selection acting directly on microsatellites, with their enhanced mutability and extended sequence motifs, leave classical signatures of a selective sweep. However, we notice that, in a broad sense, a new microsatellite allele is an indel (insertion or deletion of (a) motif(s)), and indels or even transposable element insertions have been known to be targets of selection associated with selective sweeps (51,52)."

The additional results and discussion paragraph should address the reviewer’s concerns and have greatly improved the manuscript.

Minor Concerns

1. p. 2, ln 44: The authors compare array length variation, not “array length mutations”

Fixed.

2. (throughout) Principal component analysis, not Principle component analysis.

Fixed.

3. (throughout) Strange use of colons. For the most part, the authors seem to use them like an em dash would be used.

We use colons to add information to a prior independent clause or to introduce a list after an independent clause. This recommendation comes from a popular writer’s guide by June Casagrande entitled “The best punctuation book, period”. To some extent this is a matter of preference we have elected not to change.

4. p. 6, ln 116: just curious if higher PCs (e.g., PC3 or PC4) lead to separation of AMR and SAS super populations. More importantly, how much variation do the first two PCs explain?

Interesting question; we do see that AMR and SAS populations are not well separated in PC1 and PC2. These two population do separate when we look at PC3, but only for SPS microsatellites (now referred to as EBML). AMR and SAS sill overlap quite a bit when we look at all microsatellites. It’s also surprising that EUR populations are the first to separate from the triplet of populations AMR, SAS, and EUR when we look at genetic variation across all microsatellites. The amount of variation in PCs 1-15 if fairly consistent for all microsatellites compared to EBML. Here we provide the reviewer with a summary image including additional principal component bi-plots and variation for principal components 1-15 (see attached document). We have updated the first figure of the manuscript to show the variation explained by PC1 and PC2.

5. p. 7, lns 143-146. The authors state that eSTRs are commonly found in regions subject to purifying selection, then define a sweep in terms of positive selection.

This passage was confusing. The motivation for looking at soft sweep regions is now clarified:

"The correspondence between EBML and eSTRs suggest some microsatellites contribute to differential gene expression; however, this does not imply they have adaptive potential. To infer adaptive potential we identify microsatellites in selective sweep regions of the human genome. Briefly, selective sweep occurs when strong positive selection – due to a novel allele – reduces nearby genetic variation; sweep regions have been established for six populations in the 1000 Genomes Project"

6. p. 8, ln 157+ Confused by the genomic compartments used. UTRs are technically part of exons, but not the CDS. So, (1) you might make a distinction between UTRs and full+partial exons that make up the CDS … but then why a separate “coding” compartment? … or (2) you might make a distinction between UTRs and CDS … but then why a separate “exon” compartment, since UTRs and CDS comprise exons. Is there overlap between the compartments here? – i.e., are some STRs double-counted?

Indeed exons include the UTR and CDS regions; figures 3 and 4 tabulate both to accommodate the reader’s preference. We now clarify the potential confusion in the figure 3 legend:

"Note: exonic microsatellites include CDS and UTR microsatellites based on an independent series."

The situation in figure 4c is a bit trickier. The reviewers is correct that microsatellites in exons are redundant with those in cds and utr regions. We have regenerated the figure excluding exonic microsatellites so that there is no double counting. The end of the figure legend provides a note to the reader:

"Note: exonic microsatellites (not shown) include CDS and UTR microsatellites."

7. p 11., ln 197 Reference to support the claim that hydrophobic amino acids are more likely to have deleterious effects on protein function.

Added.

8. p 14., 1st paragraph Why devote a whole paragraph of a short discussion section to the idea that we have an incomplete picture of human genetic variation. Isn’t the 1000 Genomes Project (with its 26 individual populations) a big step in the right direction? These days, the reference genome is simply a jumping off point for analyses that do make use of data sets that include much more comprehensive coverage of human genetic variation.

This paragraph doesn’t mention the 1000 Genomes Project which certainly is a big step in the right direction. The intent of the paragraph is to point out potential limitations of a single reference genome even if it is merely used as a jumping off point. Genomes that differ from the reference – particularly African genomes – may contain sequences that cannot be mapped to the reference at all. The paragraph praises the recent version of the reference genome that includes alternative sequences but suggest that this approach is unsustainable; our results alone find 22,702 alleles. Still it’s debatable whether the paragraph fits into the scope of the paper which aims to contribute to what is known about functional microsatellites. We have left it in because it does not detract from the paper and provides a transition to the limitations discussed in the last paragraph. 

Reviewer #2: 

1. What were the default parameters of the perl script the authors used for finding the repeats? Did they use a constant length cutoff, or did it change with the repeat unit? Studies have reported way more than 1.6 million STRs in humans, and knowing the exact parameters the authors used will help readers understand why they had such a small set of loci to start with.

Good question! The perl script has three key integer valued parameters: (a) merge gap; (b) max motif length; and (c) max target length. Default values for these parameters were 10, 8, and 90, respectively. The script first identifies perfect repeat motifs (up to 8bp). No impurities are allowed. Next repeats in close proximity (up to 10bp) are merged. This step introduces a degree of impurities and allows for compound repeats. We do not search for repeats in excess of 90bp. Subsequent analysis of 100bp Illumina reads requires at least 5bp of 3’ and 5’ flanking sequence; in other words, we do not search for repeats exceeding the length of illumina reads. This is a fairly severe limitation that helps explain why we begin with a small set of loci. We acknowledge this limitation in our discussion:

A key limitation of our study leads us to hypothesize that many more EBML remain undiscovered. In particular, we only investigate microsatellites arrays shorter than 100bp: a number stemming from the short read (illumina) sequencing used for the 1000 Genomes Project. Microsatellite variants exceeding 100bp are truncated rendering their true array length difficult to infer.

We would also emphasize that the script used to generate the initial list of microsatellites is previously published and is freely available: http://genotan.sourceforge.net/#_Toc324410847. 

 The limitation of our analysis to 100bp and the focus on highly pure repeats explains why we investigate a small set of loci: 1.6 million generate by the initial script. In fact, this list was pared even more to mitigate the effects of improper read mapping; essentially, we make sure repeats have unique flanking regions (see METHODS � microsatellite list generation � reduction of sequence similarity). Finally, we have also made a webserver freely available to browse the preliminary list of repeats: http://www.cagmdb.org/view_micros.php. 

2. Line 112 and other relevant places should be modified to clearly convey that they have used existing data and not done the sequencing themselves.

Line 481: All samples are freely available from the 1000 Genomes Project

Now reads: Existing data was used for this study. All samples are freely available…

Line 120: We use sequencing data from the 1000 Genomes Project

Now reads: We use existing sequencing data from the 1000 Genomes Project

Line 370: Specifically, samples were downloaded from phase 3…

Now reads: We used existing data from the 1000 Genomes Project. Specifically, samples…

3. Link to eSTRs - Why did the authors choose to use all 80,980 loci rather than the 2060 loci that showed a significant effect on expression as reported by Gymrek et al.? How many common loci are there between the 2060 eSTRs and 3984 SPS?

This is an important finding so we thank the reviewer for drawing attention to any potential confusion. We have made changes to key sentences in the results:

"The [Gymrek] study identified 2,060 significant associations (among the 80,980) which established the importance of expression STRs (eSTRs). Cross-referencing the 80,980 STRs against our 316,147 microsatellites reveals 13,259 repeats in common. We constructed a 2x2 contingency table based on classifications (eSTR and/or EBML) of the 13,259 shared repeats (Fig. 2b; right panel). Remarkably, 64 loci classify as an eSTR and EBML (Supplementary table 7); the overlap is statistically significant (p=1.53e-8; χ2 test)."

It should now be clear that we do use the 2060 loci reported by Gymrek et al. and that there are 64 common loci between the 2060 eSTRs and 3984 EBML. The revision draws better attention to the contingency table used to test statistical significance and explicitly states the type of test used (χ2 test). It’s also clear where the contingency table can be found (Fig. 2b; right panel).

4. Authors are advised to elaborate on the p value testing they did to show that 64 common out of 3984 total is significant. In particular, I would like to see if they took into account the bias towards coding regions that is introduced by whole exome sequencing. As can be seen from their study, the loci considered by Gymrek et al were predominantly proximal to genic regions. And as the current study preliminarily uses exome data, the chances of overlap are higher than what is true if whole genome is considered as background. The authors must incorporate this into calculating the significance.

This is an important question to have confidence in one of our key findings. The reviewer is suggesting that whole exome sequencing – which is bias towards coding regions – creates the illusion that eSTRs and EBML overlap significantly. 

To address this question we reiterate that analysis of the overlap between eSTRs and EBML only considers 13,259 repeats: the intersection of the 80,980 STRs from Gymrek et al and our initial list of 316,147 microsatellites. We do claim that there is statistically significant overlap between these regions (p=1.526e-8). However if we redo the analysis and exclude coding regions we still find significant overlap (p=1.248e-7). If we redo the analysis a third time excluding all exonic regions we still find significant overlap (p=1.092e-6). Contingency tables of the aforementioned results are shown here: (see document)

On one hand it’s possible to argue that the p-value is not as strong when coding sequences and exons are excluded from the analysis. On the other hand, the overlap is still highly significant despite exons being excluded even when the sequencing data is bias towards coding regions. Finally, page 9 of our manuscript concludes that EBML are over-represented in introns and under-represented in exons.

"Although introns are the majority among tested microsatellites (148,464 out of 316,147), they are over-represented among EBML (Fig. 3). On the other hand, coding microsatellites are under-represented (Fig. 3). Enrichment analysis leading to these conclusions takes into account the coverage of microsatellites among samples and is robust to sample partitioning (see methods for details)."

Thus, we emphasize two conclusions: (a) eSTRs and EBML overlap significantly even outside of exons; and (b) EBML are under-represented in coding regions and over-represented in introns. These results are only made stronger by the fact that whole exome sequencing data has a bias towards coding regions.

5. Regarding the analysis of over/under representation by repeat unit length, if a larger length cutoff was used for larger repeat unit lengths, that might explain why tetra to hexamers are underrepresented in their analyses.

Good question! Essentially the reviewer is asking if the length threshold for microsatellites used number of base pairs or number of repeated units. In case of the latter, tetra to hexamers may have been disadvantaged before enrichment analysis was done; however, this is not the case. First we submit to the reviewer a tabulation of the most common microsatellites by length (base pairs) in our initial list:

+------+-------+-------+

| nMer | bases | count |

+------+-------+-------+

| 1 | 10 | 49632 |

| 1 | 11 | 26181 |

| 1 | 12 | 18554 |

| 2 | 12 | 23942 |

| 2 | 13 | 17543 |

| 2 | 14 | 11067 |

| 3 | 15 | 9119 |

| 3 | 17 | 6909 |

| 3 | 16 | 5913 |

+------+-------+-------+

+------+-------+-------+

| nMer | bases | count |

+------+-------+-------+

| 4 | 16 | 14525 |

| 4 | 19 | 11535 |

| 4 | 17 | 10350 |

| 5 | 15 | 24042 |

| 5 | 16 | 13594 |

| 5 | 19 | 10068 |

| 6 | 18 | 7692 |

| 6 | 19 | 4768 |

| 6 | 21 | 3480 |

+------+-------+-------+

The tabulation of microsatellites shows that the most common length in base pairs for various n-mer repeats is fairly constant; thus, we did not use a larger length cutoff for larger repeat units. We also provide the 2x2 tables used for preforming χ2 test of statistical significance: (see attached document)

The messages in these tables is that SPS microsatellites (now referred to as EBML) have a disproportionately high number of 1-mer and 2-mer repeats compared to the compositions of microsatellites we used for analysis. On the contrary, EBML have a disproportionately low number of 4-mer, 5-mer, and 6-mer repeats. With the exception of 2-mer repeats, the strength of these associations leaves little doubt and the overall trend is clear: the composition of EBML shifts from over-representation of mono-nucleotide and di-nucleotide repeats to under-representation of tetra-nucleotide repeats and beyond.

6. The authors call these microsatellites super-population specific, yet there are a lot of overlaps of these loci indicating they are merely highly polymorphic. Is it really correct to call them “specific”? On a related note, what happens to the analyses if they are done only with the unique loci (~2500 loci added together).

We agree that calling these microsatellites super-population specific is potentially confusing. We have addressed this weakness is several ways. First, the abstract, author summary, and introduction include a succinct definition of SPS microsatellites (now called EBML): 

"for each EBML at least one ethnicity has genotype frequencies statistically different from the remaining four"

This should clarify that the collections of EBML for each ethnicity are permitted to overlap. We have also changed the name of these regions to ethnically-biased microsatellite loci (EBML). It is more accurate to call these regions bias rather than specific. Ethnicity has been used previously as a surrogate for the less familiar term super-population.

---

## [Decision Letter · Decision Letter 1]

31 Oct 2019

Abundance of Ethnically Biased Microsatellites in Human Gene Regions

PONE-D-19-21466R1

Dear Dr. Kinney,

We are pleased to inform you that your manuscript has been judged scientifically suitable for publication and will be formally accepted for publication once it complies with all outstanding technical requirements.

With kind regards,

Arnar Palsson, Ph.D.

Academic Editor

PLOS ONE

Additional Editor Comments (optional):

Reviewers' comments:

Reviewer's Responses to Questions

**Comments to the Author**

1. If the authors have adequately addressed your comments raised in a previous round of review and you feel that this manuscript is now acceptable for publication, you may indicate that here to bypass the “Comments to the Author” section, enter your conflict of interest statement in the “Confidential to Editor” section, and submit your "Accept" recommendation.

Reviewer #1: All comments have been addressed

Reviewer #2: All comments have been addressed

2. Is the manuscript technically sound, and do the data support the conclusions?

Reviewer #1: Yes

Reviewer #2: Yes

3. Has the statistical analysis been performed appropriately and rigorously? 

Reviewer #1: Yes

Reviewer #2: Yes

4. Have the authors made all data underlying the findings in their manuscript fully available?

Reviewer #1: Yes

Reviewer #2: Yes

5. Is the manuscript presented in an intelligible fashion and written in standard English?

Reviewer #1: Yes

Reviewer #2: Yes

6. Review Comments to the Author

Reviewer #1: Thank you for addressing my concerns effectively and presenting the changes you made in such a comprehensive and lucid manner.

Reviewer #2: (No Response)

7. PLOS authors have the option to publish the peer review history of their article (what does this mean?). If published, this will include your full peer review and any attached files.

Reviewer #1: Yes: Ryan J. Haasl

Reviewer #2: Yes: Rakesh Mishra

---

## [Editor Report · Acceptance letter]

27 Nov 2019

PONE-D-19-21466R1 

Abundance of Ethnically Biased Microsatellites in Human Gene Regions 

Dear Dr. Kinney:

I am pleased to inform you that your manuscript has been deemed suitable for publication in PLOS ONE. Congratulations! Your manuscript is now with our production department. 

With kind regards,

on behalf of

Dr. Arnar Palsson 

Academic Editor

PLOS ONE